# MapDream: Task-Driven Map Learning for Vision-Language Navigation

**Guoxin Lian** [* 1 2 ‡] **Shuo Wang** [* 1 2 ‡] **Yucheng Wang** [2 †] **Yongcai Wang** [1] **Maiyue Chen** [2] **Kaihui Wang** [2] **Bo Zhang** [2] **Zhizhong Su** [2] **Deying Li** [1] **Zhaoxin Fan** [3]

## Abstract

Vision-Language Navigation (VLN) requires agents to follow natural language instructions in partially observed 3D environments, motivating map representations that aggregate spatial context beyond local perception. However, most existing approaches rely on hand-crafted maps constructed independently of the navigation policy. We argue that maps should instead be learned representations shaped directly by navigation objectives rather than exhaustive reconstructions. Based on this insight, we propose Map-Dream, a map-in-the-loop framework that formulates map construction as autoregressive bird's-eye-view (BEV) image synthesis. The framework jointly learns map generation and action prediction, distilling environmental context into a compact three-channel BEV map that preserves only navigation-critical affordances. Supervised pre-training bootstraps a reliable mapping-to-control interface, while the autoregressive design enables end-to-end joint optimization through reinforcement fine-tuning. Experiments on R2R-CE and RxR-CE achieve state-of-the-art monocular performance, validating task-driven generative map learning. Our code and data are available at https://horizonrobotics.github.io/robot_lab/mapdream.

## 1. Introduction

Vision-Language Navigation (VLN) (Wu et al., 2024; Anderson et al., 2018; Gu et al., 2022) is a challenging task

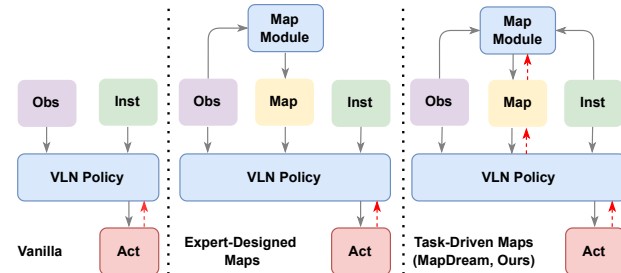

*Figure 1.* **Map-in-the-Loop Architecture.** Unlike previous approaches that either omit maps or rely on expert-designed representations, MapDream adopts a map-in-the-loop design that learns a task-driven generative map jointly with the navigation policy. Red dashed arrows denote training-time gradient flow from navigation objectives, illustrating how the learned map representation is directly shaped by downstream tasks. Abbreviations: Obs denotes observations, Inst instructions, and Act actions.

in the field of embodied artificial intelligence that requires agents to ground natural language instructions into coherent action sequences within complex environments. A central difficulty of VLN is partial observability. Agents perceive the environment only through local, sequential observations, which limits their understanding of the space. During navigation, they must reason over incomplete and progressively revealed spatial information. As a result, in current VLN pipelines, aggregating past observations into a persistent spatial state is a standard and integral component. Hence, most existing VLN methods (Chen et al., 2022; Zhang et al., 2025b; Wang et al., 2025e; Zeng et al., 2025) incorporate map representations to provide spatial context for decision making.

Many existing map representation methods, such as topological graphs, BEV representations (An et al., 2022), or grid-based memories (Wang et al., 2023b), help mitigate partial observability and provide spatial abstractions useful for navigation. These approaches enhance navigation performance by offering spatial context, particularly under partial observability, and improve decision-making (Georgakis et al., 2022; An et al., 2024; Huang et al., 2022; Cai et al., 2024). The strength of these methods lies in providing valuable environmental information through maps, boosting overall navigation performance. However, most of these

*Equal contribution †Project Leader ‡This work was done while Guoxin Lian and Shuo Wang were Research Interns with Horizon Robotics. ¹Renmin University of China ²Horizon Robotics ³Beijing Advanced Innovation Center for Future Blockchain and Privacy Computing. Correspondence to: Yongcai Wang <ycw@ruc.edu.cn>, Zhaoxin Fan <zhaoxinf@buaa.edu.cn>.

*Proceedings of the 43rd International Conference on Machine Learning*, Seoul, South Korea. PMLR 306, 2026. Copyright 2026 by the author(s).

maps are constructed independently of the decision-making process and are consumed by the policy as fixed inputs.

The limitation of this approach is that map representations typically remain outside the learning loop that governs navigation behavior, preventing them from being refined through learning to align with task objectives (see Fig 1). Since these maps are not directly shaped by task-driven learning signals, they cannot be adjusted during training to align with the semantics of instructions or the needs of the navigation policy, leading to a mismatch between the spatial information encoded in the map and the decision-making requirements of the navigation policy. This mismatch highlights that most existing maps are designed by experts, rather than being learned end-to-end from navigation objectives. Motivated by the above analysis, we therefore argue that effective map representations for VLN should be learned as part of the decision-making process and optimized for navigation objectives. Under this task-driven formulation, maps need not encode the full state of the environment, but only a compact spatial representation sufficient for navigation decisions.

Based on this insight, we propose MapDream, a framework that unifies spatial representation learning and decision making. The proposed method is built upon three core components: a map-in-the-loop architecture, supervised pre-training, and reinforcement fine-tuning. First, the map-in-the-loop architecture comprises a task-driven map module and a VLN policy, where BEV maps are autoregressively generated from egocentric observation histories and language instructions and then provided to the policy as structured spatial context for multi-step action prediction. Second, a supervised pre-training stage constructs task-driven BEV supervision and trains both the map generator and the policy to establish a reliable mapping-to-control interface, constraining the representations to fixed resolutions and token budgets before downstream task-level optimization. Finally, joint reinforcement fine-tuning is performed under a unified navigation objective, using multi-source rewards, group-based rollout optimization, and relative-advantage objectives to directly shape both components toward encoding navigation-critical information rather than mere geometric reconstruction. We demonstrate the effectiveness of this approach by achieving state-of-the-art performance on the standard VLN benchmarks R2R-CE and RxR-CE under the monocular setting. The framework also exhibits strong generalization to unseen environments, highlighting the practical value of task-driven map representations for real-world navigation tasks.

Our main contributions are:

- We first introduce a task-driven perspective on map representations for VLN, reframing maps as representations shaped by downstream navigation objectives rather than fixed by expert design.

- We present MapDream, which formulates map construction as an autoregressive generative process and enables joint optimization of the map module and navigation policy under reinforcement learning.

- We achieve state-of-the-art results on the standard VLN benchmarks R2R-CE and RxR-CE under the monocular setting, with strong generalization to unseen environments, validating task-driven map representations as an effective paradigm.

## 2. Related Works

### 2.1. Vision Language Navigation

Large-scale pretrained vision-language models (VLMs) have recently become a strong foundation for VLN, substantially improving visual grounding and language understanding through large-scale multimodal pretraining (Bai et al., 2025b; Liu et al., 2024). Building on such pretrained VLMs, recent VLA-based VLN systems couple perception, language, and control to directly predict actions from monocular RGB streams and instruction inputs (Liu et al., 2025a; Zhang et al., 2024a; Cheng et al., 2024; Wei et al., 2025; Wang et al., 2023a), yielding substantial gains in generalization and transferability. Recent VLN research has further enhanced VLA-based navigation by injecting explicit reasoning signals and richer world priors. CoT-style supervision and progress-aware reasoning encourage models to internalize structured reasoning processes or instruction-progress states to support decision making (Wang et al., 2025b;d; Liu et al., 2025b; Dai et al., 2026). Other lines of work address partial observability by enriching visual context, either through hallucinated panoramic cues from monocular streams (Wang et al., 2025c), or by introducing a dual implicit neural memory that separately models spatial-geometric and visual-semantic information as compact, fixed-size representations (Zeng et al., 2025). Map-based VLA systems further incorporate structured spatial states in different forms, such as MapNav (Zhang et al., 2025b), which employs annotated semantic maps to replace historical frames, and Dynam3D (Wang et al., 2025e), which introduces dynamically updated layered 3D tokens as visual inputs. MapDream departs from these approaches by learning a compact BEV representation via generative map construction and integrating it into the VLN policy for end-to-end optimization.

### 2.2. Map Representation for Embodied Navigation

Map representations provide spatial context beyond local perception and have long been central to embodied navigation. Classical navigation pipelines build explicit metric, semantic, or topological maps through external mapping modules and couple them to planners or hierarchical con-

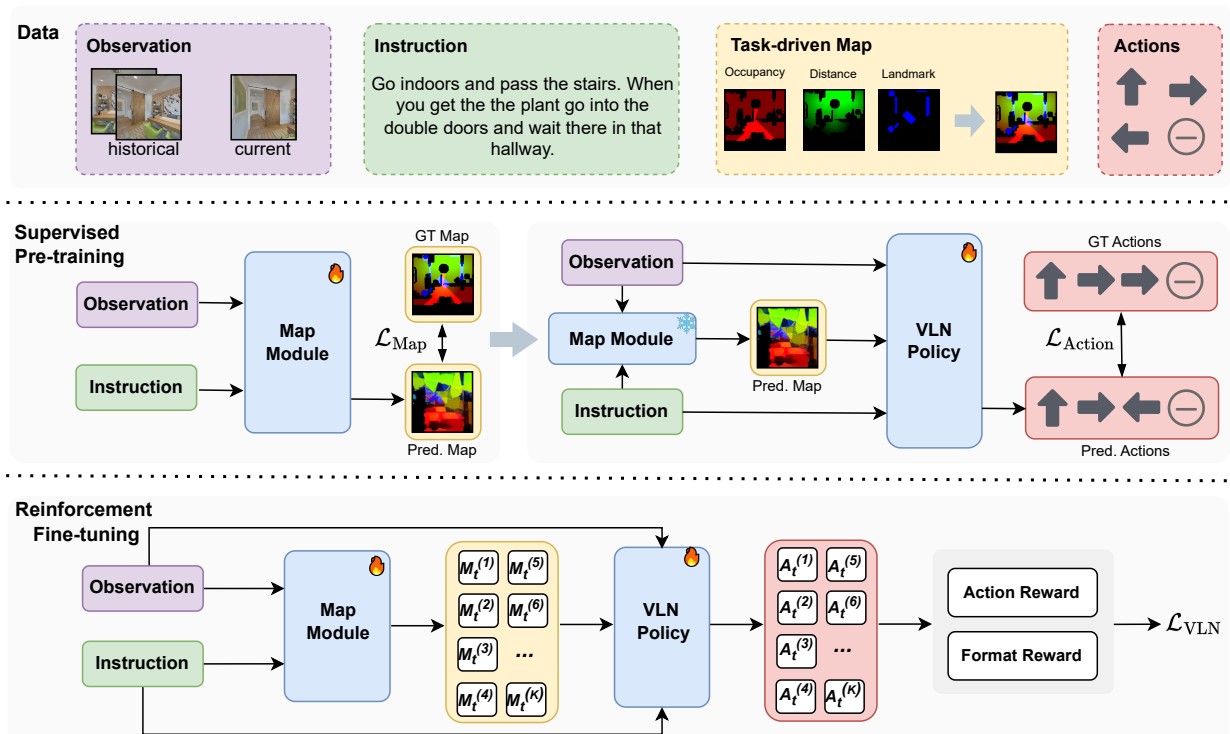

*Figure 2.* **Overview of the MapDream Framework.** The diagram shows the two-stage optimization of MapDream. Stage 1 learns structured task-driven maps from visual observations and language instructions for initialization and supervised policy training. Stage 2 jointly optimizes the map module and VLN policy through reinforcement learning under a unified navigation objective, allowing the map to be shaped by downstream tasks.

trollers (Huang et al., 2022; Li et al., 2025; Chiang et al., 2024; Wang et al., 2025a; 2024a). These formulations treat map construction as a structured prediction or reconstruction problem that emphasizes completeness and interpretability, making them well-suited for explicit mapping pipelines but less aligned with task-driven abstractions that directly support downstream decision making. Within the VLN setting, prior work has explored map-like representations as structured top-down interfaces for navigation, including multi-modal BEV pre-training (An et al., 2022), egocentric grid memory maps (Wang et al., 2023b), BEV scene graphs (Liu et al., 2023), and language-conditioned egocentric semantic map prediction (Georgakis et al., 2022). While effective, these representations are typically constructed using fixed design choices or auxiliary objectives, and remain weakly coupled to the downstream action policy. In contrast, Map-Dream learns map representations in a task-driven manner, where the content and structure of the map are shaped by downstream navigation objectives rather than fixed design choices.

## 2.3. Map Construction as Image Synthesis

Viewing maps as compact decision-oriented representations removes the need for exhaustive environment reconstruc-tion and naturally connects map construction to recent advances in generative modeling, which demonstrate that complex visual structures can be synthesized from partial and high-level conditioning signals. Both diffusion-based and autoregressive models exhibit strong controllability and semantic alignment when generating images conditioned on text or multimodal inputs, such as instruction-guided image editing and multimodal image manipulation (Rombach et al., 2022; Chen et al., 2023a; Brooks et al., 2023; Bai et al., 2025a). These successes establish image synthesis as a powerful paradigm for generating structured visual representations. Autoregressive multimodal models such as Janus-Pro (Chen et al., 2025) further unify visual and textual generation within a single generative framework, enabling flexible conditioning over mixed-modality inputs and sequential generation over heterogeneous tokens. However, such models typically assume same-view generation and condition on a single image, making them ill-suited for VLN map construction, which requires cross-view synthesis from egocentric observations to bird's-eye-view (BEV) representations, cross-domain generation from natural images to abstract spatial maps, and conditioning on multi-frame observation histories under partial observability. MapDream formulates map construction as image synthesis for embod-ied navigation, generating compact BEV images from multi-

frame egocentric observations and language instructions that are sufficient to directly support navigation decisions.

# 3. Method

## 3.1. Overview

MapDream is built upon three core components, as shown in Fig. 2. Specifically, it features (1) a two-module system composed of a task-driven map module and a VLN policy for spatial representation learning and action prediction; (2) a supervised pre-training stage that establishes a reliable mapping-to-control interface for structured BEV representations under fixed resolution and token budgets; and (3) a joint reinforcement fine-tuning stage under a unified navigation objective that directly shapes both the map representation and the policy.

## 3.2. Map-in-the-Loop Architecture

MapDream adopts a two-component architecture consisting of a task-driven map module and a VLN policy that operate throughout both training stages. At each time step $t$, the map module receives an egocentric observation history $O_t$, the current frame $o_t$, and the instruction $I$, and autoregressively generates a multi-channel BEV representation $M_t$ that summarizes navigationally relevant spatial context. This BEV map serves as an explicit spatial interface between perception and control and is provided to the VLN policy together with $O_t$, $o_t$, and $I$.

The VLN policy consumes the map and predicts a sequence of $N$ future actions $a_{t:t+N-1}$, enabling multi-step planning under partial observability. Across both supervised pre-training and reinforcement fine-tuning, the two components are optimized either with separate objectives or under a unified navigation reward, while remaining connected through the BEV representation, which functions as the sole spatial representation passed between modules.

By explicitly decoupling spatial abstraction from action prediction through the BEV interface, MapDream allows the map module to focus on constructing navigation-relevant representations while the policy concentrates on decision making.

## 3.3. Supervised Pre-training

In Stage 1, we train the map module and the VLN policy with separate supervised objectives to provide a stable initialization before reinforcement fine-tuning. This stage focuses on constructing task-driven BEV supervision and optimizing the two components with separate losses. It consists of three parts: task-driven map supervision, pre-training the map module, and pre-training the VLN policy.

### 3.3.1. MAP SUPERVISION

We adopt lightweight ground-truth map signals during supervised pre-training to encode navigation-critical cues; this design is not exclusive, and alternative compact variants are possible. In MapDream, the map supervision is represented as a three-channel BEV image consisting of Occupancy, Distance, and Landmark maps, reflecting common spatial cues exploited in embodied navigation, where traversability, goal-directed geometry, and stable semantic anchors are crucial for long-horizon planning and instruction grounding.

The Occupancy map uses $\{0, 128, 255\}$ to represent observed impassable, unobserved, and observed traversable regions, respectively. The Distance map provides a dense relative geodesic-to-goal signal centered at the agent, normalized and truncated to $[0, 255]$, while the Landmark map highlights static scene objects referenced in natural language instructions. Together, these channels form a BEV representation that captures complementary aspects of geometry, goal structure, and semantic grounding beyond the current field of view.

### 3.3.2. PRE-TRAINING THE MAP MODULE

Using the generated task-driven maps as supervision, we train an autoregressive model to predict BEV maps from egocentric observations and language instructions:

$$M_t = G(O_t, o_t, I), \tag{1}$$

where $G$ denotes the map module. The generation process is supervised with a reconstruction objective,

$$\mathcal{L}_{\text{Map}} = -\sum_t \log p_\theta(\hat{y}^t \mid \hat{y}^{<t}, O_t, o_t, I), \tag{2}$$

which encourages spatially consistent and task-relevant BEV predictions.

### 3.3.3. PRE-TRAINING THE VLN POLICY

The VLN policy is trained to predict multi-step action sequences conditioned on the predicted maps and visual–language context by minimizing a cross-entropy loss,

$$\mathcal{L}_{\text{Action}} = -\log \pi\left(a_{t:t+N-1}^* \mid O_t, o_t, M_t, I\right). \tag{3}$$

This objective equips the policy with multi-step decision capability and provides a strong initialization for subsequent reinforcement fine-tuning.

## 3.4. Reinforcement Fine-tuning

In Stage 2, we jointly fine-tune the map module and the VLN policy under a unified navigation objective to directly optimize task performance. Optimization is guided by two reward components—an action reward and a format

reward—and is carried out using Group Relative Policy Optimization (GRPO) (Shao et al., 2024).

**Action Reward.** The action reward enforces stepwise correctness in the predicted action sequence by rewarding only the longest correct prefix. Once an incorrect action appears, no further reward is assigned, reflecting the sequential credit assignment nature of navigation. It is defined as:

$$r_{\text{act}} = \sum_{i=0}^{N-1} \prod_{j=0}^{i} \mathbb{1}[a_{t+j} = a_{t+j}^*]. \tag{4}$$

**Format Reward.** The format reward ensures that the predicted action sequence satisfies the required syntactical structure:

$$r_{\text{fmt}} = \begin{cases} 1, & \text{if } a_{t:t+N-1} \text{ is valid,} \\ 0, & \text{otherwise.} \end{cases} \tag{5}$$

**Total Reward.** The total reinforcement signal is defined as $r_{\text{total}} = r_{\text{act}} + r_{\text{fmt}}$.

**Rollout.** During each rollout of length $K$, the two components operate as a coupled process, producing a sequence of predicted maps and action proposals $\{M_t\}_{t=0}^{K-1}$ and $\{a_{t:t+N-1}\}_{t=0}^{K-1}$.

**Optimization Objective.** We sample a group of candidate rollouts indexed by $k$ and compute relative advantages within the group. The optimization objective follows a GRPO-style formulation:

$$\mathcal{L}_{\text{VLN}} = -\mathbb{E}_k \left[ \min \begin{pmatrix} \rho(k)\, A(k), \\ \text{clip}\big(\rho(k),\, 1-\epsilon,\, 1+\epsilon\big)\, A(k) \end{pmatrix} \right], \tag{6}$$

where the relative advantage ratio $\rho(k)$ is defined as

$$\rho(k) = \frac{\pi_\theta^{\text{nav}}\big(a_{t:t+N-1}^{(k)} \,\big|\, M_t^{(k)}, O_t, o_t, I\big)}{\pi_{\theta_{\text{old}}}^{\text{nav}}\big(a_{t:t+N-1}^{(k)} \,\big|\, M_t^{(k)}, O_t, o_t, I\big)} \\ \cdot \frac{p_\phi^{\text{bev}}\big(M_t^{(k)} \,\big|\, O_t, o_t, I\big)}{p_{\phi_{\text{old}}}^{\text{bev}}\big(M_t^{(k)} \,\big|\, O_t, o_t, I\big)}, \tag{7}$$

where $\pi_\theta^{\text{nav}}$ and $p_\phi^{\text{bev}}$ denote the VLN policy and map module, respectively. $(\theta, \phi)$ and $(\theta_{\text{old}}, \phi_{\text{old}})$ indicate current and reference parameters, and $A(k)$ is the normalized advantage at step $k$.

# 4. Experiments

## 4.1. Experiment Setup

### 4.1.1. EXPERIMENTAL ENVIRONMENTS

We evaluate our method on the widely adopted continuous-environment VLN benchmarks R2R-CE (Krantz et al., 2020)

and RxR-CE (Ku et al., 2020). Results are reported on the validation-unseen splits to assess generalization to novel environments. RxR is more challenging than R2R, featuring substantially longer trajectories and multilingual, fine-grained instructions that demand strong global spatial reasoning. We focus on the continuous-environment (CE) protocol because continuous control introduces fine motion granularity and realistic noise, making navigation sensitive to small geometric deviations.

### 4.1.2. METRICS

We adopt the standard VLN evaluation protocol (Krantz et al., 2020; Ku et al., 2020) to assess navigation performance using success rate (SR), oracle success rate (OSR), success weighted by path length (SPL), and navigation error (NE). We primarily report SR and SPL, which capture task completion and path efficiency, respectively.

## 4.2. Implementation Details

### 4.2.1. DATASET COLLECTION

We leverage continuous VLN simulators from R2R-CE and RxR-CE to construct training data. Their annotated trajectories are converted into step-level samples, yielding 1200K state-action pairs. To support Stage 1 training, we generate BEV maps for these datasets. We follow standard egocentric BEV configurations used in recent VLN systems (Chen et al., 2022; An et al., 2022), ensuring compatibility with indoor navigation tasks. The Occupancy and Distance channels are derived from the Habitat simulator, while the Landmark channel is annotated using 3D referential grounding labels from IRef-VLA (Zhang et al., 2025a). These BEV representations serve as the initial input to MapDream for map generation and policy learning. Additionally, we generate 500K non-oracle samples through exploratory rollouts in the training environments, improving robustness to out-of-distribution states and enhancing generalization across diverse scenarios.

### 4.2.2. TRAINING DETAILS

The training of MapDream follows a two-stage procedure and is conducted on 8 NVIDIA H20 GPUs. Stage 1 performs supervised pre-training of both the map module and the VLN policy, while Stage 2 jointly fine-tunes them with reinforcement learning. Stage 1 runs for two epochs and takes approximately 60 hours, and Stage 2 performs 2000 RL steps and takes approximately 10 hours.

In Stage 1, we train Janus-Pro as the map module using ground-truth BEV maps. Janus-Pro is trained for one epoch with a batch size of 40 and a learning rate of $1 \times 10^{-4}$ in a supervised pre-training manner following Janus-Pro-R1 (Pan et al., 2026), extended to support multi-image and

*Table 1.* Comparison of different methods on the R2R-CE Val-Unseen and RxR-CE Val-Unseen splits. Observations used include single RGB camera (S.RGB), depth sensor (Depth), panoramic view (Pano.) and map representation (Map). † indicates methods without using LLMs.

| Method | Venue | Observation | | | Map | R2R-CE Val-Unseen | | | | RxR-CE Val-Unseen | | |
|---|---|---|---|---|---|---|---|---|---|---|---|---|
| | | S.RGB | Depth | Pano. | | NE ↓ | OSR ↑ | SR ↑ | SPL ↑ | NE ↓ | SR ↑ | SPL ↑ |
| GridMM†(Wang et al., 2023b) | ICCV2023 | | ✓ | ✓ | ✓ | 5.11 | 61.0 | 49.0 | 41.0 | – | – | – |
| ETPNav†(An et al., 2024) | TPAMI2024 | | ✓ | ✓ | ✓ | 4.71 | 65.0 | 57.0 | 49.0 | 5.64 | 54.7 | 44.8 |
| BEVBert†(An et al., 2022) | ICCV2023 | | ✓ | ✓ | ✓ | 4.57 | 67.0 | 59.0 | 50.0 | – | – | – |
| BEVBert-FSTTA†(Gao et al., 2023) | ICML2024 | | ✓ | ✓ | ✓ | 4.39 | 65.0 | 60.0 | 51.0 | – | – | – |
| BEVBert-FEEDTTA†(Kim et al., 2025) | ICML2025 | | ✓ | ✓ | ✓ | 4.33 | 69.0 | 61.0 | 50.0 | – | – | – |
| NaVid-4D (Liu et al., 2025a) | ICRA2025 | ✓ | ✓ | | | 5.99 | 55.7 | 43.8 | 37.1 | – | – | – |
| sim2real†(Wang et al., 2024b) | CoRL2024 | ✓ | ✓ | | | 5.95 | 55.8 | 44.9 | 30.4 | – | – | – |
| NavMorph†(Yao et al., 2025) | ICCV2025 | ✓ | ✓ | | | 5.75 | 56.9 | 47.9 | 33.2 | 8.85 | 30.7 | 22.8 |
| CM2†(Georgakis et al., 2022) | CVPR2022 | ✓ | ✓ | | ✓ | 7.02 | 41.0 | 34.0 | 27.0 | – | – | – |
| WS-MGMap†(Chen et al., 2022) | NeurIPS2022 | ✓ | ✓ | | ✓ | 6.28 | 47.0 | 38.0 | 34.0 | – | – | – |
| NaVid (Zhang et al., 2024b) | RSS2024 | ✓ | | | | 5.47 | 49.1 | 37.4 | 35.9 | – | – | – |
| Uni-NaVid(Zhang et al., 2024a) | RSS2025 | ✓ | | | | 5.58 | 53.3 | 47.0 | 42.7 | 6.24 | 48.7 | 40.9 |
| NaVILA(Cheng et al., 2024) | RSS2025 | ✓ | | | | 5.22 | 62.5 | 54.0 | 49.0 | 6.77 | 49.3 | 44.0 |
| Aux-Think (Wang et al., 2025b) | NeurIPS2025 | ✓ | | | | 6.08 | 60.0 | 54.8 | 46.9 | – | – | – |
| MonoDream(Wang et al., 2025c) | AAAI2025 | ✓ | | | | 5.45 | 61.5 | 55.8 | 49.1 | – | – | – |
| MapNav (Zhang et al., 2025b) | ACL2025 | ✓ | | | ✓ | 4.93 | 53.0 | 39.7 | 37.2 | 7.62 | 32.6 | 27.7 |
| Dynam3D (Wang et al., 2025e) | NeurIPS2025 | ✓ | | | ✓ | 5.34 | 62.1 | 52.9 | 45.7 | – | – | – |
| MapDream (Ours) | – | | ✓ | | ✓ | **4.59** | **64.4** | **59.8** | **54.4** | **4.96** | **59.4** | **49.2** |

text inputs for cross-modal BEV generation. After SPT, Janus-Pro is frozen to provide stable task-driven spatial representations.

The generated BEV maps are then used to condition the VLN policy. The policy is initialized from pretrained NVILA-2B weights and trained with a mixture of oracle expert trajectories and DAgger-collected data. We optimize the policy with cross-entropy loss over the next three predicted action steps at each time step using a learning rate of $1 \times 10^{-5}$, providing a supervised cold-start for map-conditioned navigation.

In Stage 2, we apply GRPO optimization with a rollout size of 8, a clipping parameter of 0.28, and a KL coefficient of 0.0. The model is trained for 2000 steps with a learning rate of $1 \times 10^{-6}$, jointly fine-tuning both the map module and the VLN policy. Reinforcement signals consist of action and format rewards, enabling credit assignment to both spatial representations and policy decisions.

*Table 2.* Unseen-Dataset generalization performance on the RxR-CE Val-Unseen split. All results are obtained only training on the R2R-CE training set.

| Method | RxR Val-Unseen | | | |
|---|---|---|---|---|
| | NE ↓ | OSR ↑ | SR ↑ | SPL ↑ |
| CM2(Georgakis et al., 2022) | 8.98 | 25.3 | 14.4 | 9.2 |
| WS-MGMap(Chen et al., 2022) | 9.83 | 29.8 | 15.0 | 12.1 |
| A$^2$NAV(Chen et al., 2023b) | - | - | 16.8 | 6.3 |
| NaVid(Zhang et al., 2024b) | **8.41** | 34.5 | 23.8 | 21.2 |
| MonoDream(Wang et al., 2025c) | 8.57 | 35.9 | 25.1 | 21.6 |
| sim2real(Wang et al., 2024b) | 8.79 | 36.7 | 25.5 | 21.2 |
| MapDream (Ours) | 8.73 | **38.4** | **27.8** | **23.3** |

## 4.3. Comparison with State-of-the-Art Methods

We compare MapDream with state-of-the-art methods on the R2R-CE and RxR-CE benchmarks under a single RGB camera (monocular) observation setting. As shown in Table 1, MapDream achieves the best overall performance among monocular approaches on both datasets, with the highest SR and SPL while reducing navigation error. It reaches 59.8 SR and 54.4 SPL on R2R-CE val-unseen and 59.4 SR and 49.2 SPL on RxR-CE, establishing new state-of-the-art results in this regime. Despite relying only on monocular inputs, MapDream surpasses panoramic-based methods in terms of SPL, which measures both success and path efficiency, indicating that its successful trajectories closely follow the reference demonstration paths.

Table 2 further reports zero-shot generalization to RxR-CE when training only on R2R-CE. MapDream again attains the strongest SR and SPL among all competitors, suggesting that the learned maps capture transferable navigation-relevant structure rather than dataset-specific cues.

Across all settings, MapDream improves both success rate and path efficiency, which we attribute to its task-driven generative maps that are refined through two-stage optimization and reinforcement fine-tuning. These results empirically validate that learning spatial abstractions under navigation objectives leads to more robust decision making in continuous environments.

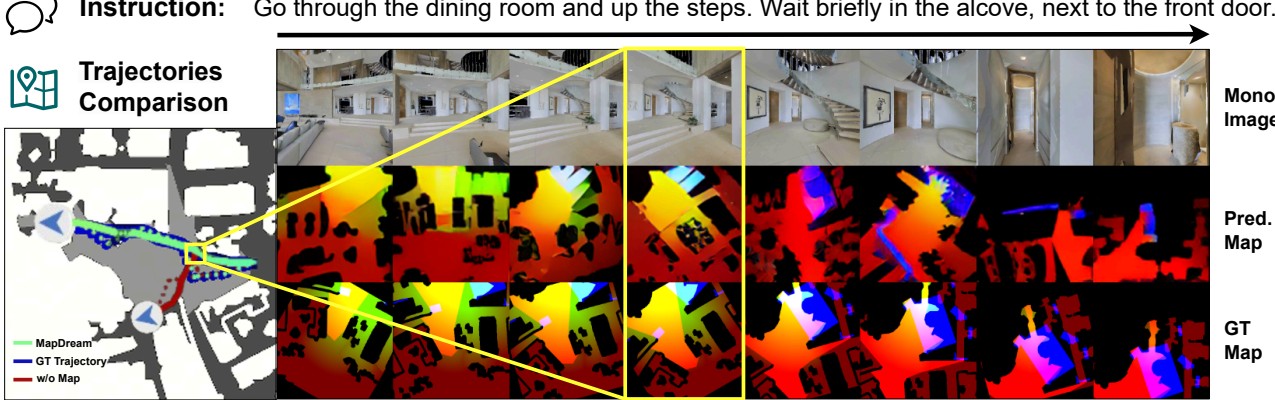

*Figure 3.* **Qualitative navigation example illustrating the effect of task-driven maps in MapDream.** (Left) Trajectory comparison shows that MapDream (green) closely follows the ground-truth path (blue), while the VLN policy without maps deviates (red). (Right) Conditioned on monocular observations, MapDream generates task-driven BEV maps that retain navigation-critical spatial cues such as occupancy, distance, and landmarks. These maps provide compact task-driven abstractions for decision-making. The yellow-highlighted region marks a spatial decision point where the no-map policy deviates, whereas MapDream selects the correct path using map information.

### 4.4. Qualitative Analysis

To better understand how task-driven maps benefit navigation, we visualize trajectories and intermediate representations in MapDream (see Fig. 3). As shown on the left, MapDream produces navigation paths that closely align with ground-truth trajectories, whereas the VLN policy without maps deviates significantly. This suggests that the maps provide actionable spatial cues for decision-making during long-horizon navigation. On the right, we examine the predicted BEV maps generated from monocular observations. The maps retain navigation-relevant spatial structures, including occupancy, distance, and landmarks, and resemble the ground-truth layouts while remaining compact and task-driven. These results indicate that the learned maps serve as effective abstractions of the environment, supporting accurate navigation without requiring full scene reconstruction.

### 4.5. Ablation Study

We conduct three ablation studies on R2R-CE that jointly probe MapDream along complementary design dimensions: optimization strategy, robustness to map initialization, and representation capacity. Concretely, we analyze the effect of two-stage training, the sensitivity of reinforcement fine-tuning to different channel initializations, and the trade-off between BEV map compactness and cold-start navigation performance.

#### 4.5.1. TWO-STAGE TRAINING

We evaluate the effect of two-stage training in MapDream by comparing three configurations: a baseline VLN policy without maps, the map-conditioned model after Stage 1 supervised pre-training, and the full two-stage system with additional Stage 2 reinforcement fine-tuning.

As shown in Table 3, SPT corresponds to Stage 1 supervised pre-training with map conditioning, while RFT denotes Stage 2 reinforcement fine-tuning that jointly updates the map module and the VLN policy. Introducing Stage 1 yields consistent improvements across all metrics over the baseline, demonstrating that generative task-driven maps provide useful spatial abstractions for instruction-following navigation. Stage 2 further boosts SR and SPL while reducing NE, indicating that reinforcement learning aligns the learned maps with downstream navigation behavior. Together, the two stages contribute complementary gains for long-horizon navigation.

*Table 3.* Effect of staged learning on R2R-CE val-unseen.

| Map | SPT | RFT | NE↓ | OSR↑ | SR↑ | SPL↑ |
|---|---|---|---|---|---|---|
| - | - | - | 7.67 | 45.8 | 37.7 | 32.1 |
| ✓ | ✓ | - | 7.03 | 48.0 | 42.2 | 36.5 |
| ✓ | ✓ | ✓ | **6.35** | **51.3** | **45.6** | **40.5** |

#### 4.5.2. REINFORCEMENT FINE-TUNING UNDER DIFFERENT CHANNEL INITIALIZATIONS

To further examine the effect of reinforcement fine-tuning under different channel initializations, Table 4 reports results for maps initialized with only the Occupancy, Distance, or Landmark channel, as well as their full combination. Reinforcement learning consistently improves all variants, with SR gains of +3.4 (All), +5.5 (Distance), +4.1 (Landmark), and +4.2 (Occupancy), accompanied by increases in SPL (+3.6-5.5), OSR, and reduced NE.

Although different channel choices yield different starting

performance after supervised pretraining, reinforcement fine-tuning narrows these gaps, bringing all variants to similar final SR (43.6-45.6) and SPL (39.4-40.5). This indicates that Stage 2 is largely insensitive to the specific channel used for initialization. Overall, these results support MapDream's premise that generative, task-driven BEV maps can be effectively aligned with navigation objectives through reinforcement learning.

*Table 4.* Effect of Reinforcement Fine-tuning under Different Channel Initializations.

| Channel | SPT | RFT | NE↓ | OSR↑ | SR↑ | SPL↑ |
|---|---|---|---|---|---|---|
| All | ✓ | - | 7.03 | 48.0 | 42.2 | 36.5 |
| | ✓ | ✓ | **6.35** | **51.3** | **45.6** | **40.5** |
| Distance | ✓ | - | 7.18 | 45.4 | 39.4 | 34.5 |
| | ✓ | ✓ | **6.49** | **51.2** | **44.9** | **40.0** |
| Landmark | ✓ | - | 7.32 | 48.7 | 40.4 | 35.3 |
| | ✓ | ✓ | **6.60** | **51.9** | **44.5** | **39.7** |
| Occupancy | ✓ | - | 7.39 | 45.4 | 39.4 | 34.8 |
| | ✓ | ✓ | **6.67** | **49.6** | **43.6** | **39.4** |

### 4.5.3. BEV MAP RESOLUTION AND TOKEN BUDGET

We study the impact of BEV map resolution and token budget in the map generator under Stage 1. As shown in Table 5, increasing map size improves reconstruction fidelity but brings only marginal gains in navigation performance. Notably, the most compact configuration attains a comparable success rate to the largest model (42.2 vs. 43.1 SR), indicating that dense geometric reconstruction is unnecessary for effective map-conditioned navigation. Instead, generative task-driven maps primarily serve as actionable spatial abstractions for policy bootstrapping.

Reducing map size also substantially improves efficiency. In particular, inference latency per decision step drops from 12.7 s to 1.3 s, making compact maps far more suitable for real-time continuous control. This highlights a favorable accuracy-efficiency tradeoff and suggests that MapDream naturally operates in a low-resolution, low-token regime without sacrificing navigation performance.

*Table 5.* Effect of BEV token capacity on R2R-CE val-unseen.

| Resolution | Tokens | NE ↓ | OSR ↑ | SR ↑ | SPL ↑ | Time(s) |
|---|---|---|---|---|---|---|
| $384 \times 384$ | 576 | **6.92** | 48.2 | **43.1** | **38.7** | 12.7 |
| $192 \times 192$ | 144 | 7.12 | 48.0 | 42.7 | 37.7 | 4.5 |
| $96 \times 96$ | 36 | 7.03 | 48.0 | 42.2 | 36.5 | **1.3** |

### 4.6. Real-world Generalization

To assess real-world generalization, we deploy MapDream on a Unitree G1 humanoid equipped with a forward-facing

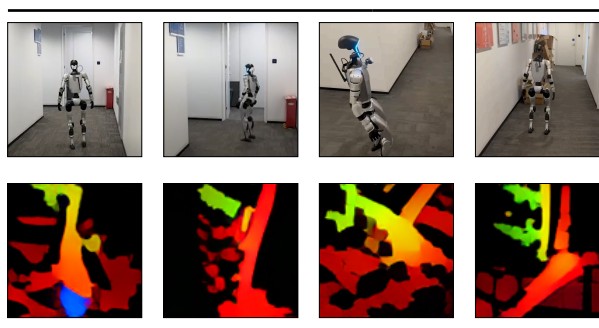

**Instruction:** Walk straight down the hallway and turn left at the open doorway. Continue down the corridor and turn left at the end, then stop in front of the first cardboard box.

*Figure 4.* **Real-World Navigation with Task-Driven Maps.** Real-world deployment of MapDream on a humanoid platform. Given monocular input and a natural language instruction, MapDream constructs robot-centric task-driven BEV maps from the forward-facing viewpoint. The maps capture navigation-relevant spatial affordances over time and guide the robot to execute accurate long-horizon navigation in real indoor environments.

RGB camera. The robot receives monocular observations and executes navigation actions directly from live inputs. As shown in Fig. 4, MapDream generates task-driven maps that evolve with the robot's motion and encode navigation-relevant spatial affordances, allowing the robot to follow long-horizon language instructions successfully in real indoor environments. Notably, the model is trained only on the R2R-CE and RxR-CE simulators, yet transfers in a zero-shot manner to real-world, previously unseen indoor scenes.

## 5. Conclusion

In this work, we introduce MapDream, a task-driven framework designed to improve VLN by optimizing the generation of map representations. Unlike traditional VLN methods, which either omit maps or rely on hand-crafted designs, MapDream rethinks map construction as an autoregressive generative process. By jointly optimizing both map generation and navigation policies through a two-stage training approach, supervised pre-training followed by reinforcement fine-tuning, MapDream enables agents to learn compact, task-relevant map representations that evolve with the navigation policy and remain effective across different initialization choices and representation capacities. This synergistic framework significantly improves navigation success and efficiency on established VLN benchmarks, such as R2R-CE and RxR-CE, while demonstrating strong generalization to unseen environments. Our findings show that task-driven map representations are key to improving the performance and scalability of VLN systems. More broadly, rethinking maps through a task-driven lens may offer a scalable paradigm for future embodied AI.

**Limitations.** MapDream currently maintains a local egocen-

tric BEV representation that aggregates recent observations into a broader spatial context, which substantially improves over frame-level reasoning but does not explicitly model a persistent global map over arbitrarily long horizons. In addition, temporal consistency of the generated maps across timesteps is not explicitly enforced. Extending the framework to incorporate long-term global spatial memory and temporally coherent map representations is a promising direction for future work.

## Acknowledgements

This work was supported by the New Generation Artificial Intelligence-National Science and Technology Major Project (2025ZD0122603). It was also supported by the Postdoctoral Fellowship Program and China Postdoctoral Science Foundation under Grant No. 2024M764093 and Grant No. BX20250485, the Beijing Natural Science Foundation under Grant No. 4254100, and by Beijing Advanced Innovation Center for Future Blockchain and Privacy Computing.

## Impact Statement

This paper presents work whose goal is to advance the field of machine learning. There are many potential societal consequences of our work, none of which we feel must be specifically highlighted here.

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
