# OpenReview forum: "MapDream: Task-Driven Map Learning for Vision-Language Navigation"
_ICML.cc/2026/Conference — ICML 2026 regular_

### Official Review · Reviewer_xqed · 2026-03-11

**Soundness:** 3
**Presentation:** 2
**Significance:** 2
**Originality:** 3
**Overall Recommendation:** 4
**Confidence:** 3

**Summary:**

This paper studies map learning for vision-language navigation in continuous environments. The paper argues that many existing VLN methods either do not use maps or rely on hand-crafted map representations that are only weakly coupled to downstream decision making. To address this, they propose MapDream, which formulates map construction as autoregressive BEV image generation from egocentric monocular observation histories and language instructions. The framework contains a map module and a VLN policy, and is trained in two stages. In the first stage, the model is supervised using a compact three-channel BEV target consisting of occupancy, distance, and landmark maps. In the second stage, the map module and policy are jointly fine-tuned with reinforcement learning under a unified navigation objective, with the goal of making the learned map representation better aligned with navigation performance.

**Compliance With Llm Reviewing Policy:**

Affirmed.

**Final Justification:**

Thanks to the authors for the detailed rebuttal and the new experiments. I believe most of my concerns have been addressed.

I recommend incorporating these materials into the final version (main paper or appendix) to make the contribution more complete.

I will increase my score for weak acceptance.

**Key Questions For Authors:**

1. The paper frames the learned representation as a “task-driven” map, but Stage 1 relies on manually designed BEV supervision with occupancy, distance, and landmark channels. Could the authors clarify in what sense the final representation should be considered genuinely task-driven, rather than largely shaped by these predefined targets?

2. The current experiments do not fully isolate the source of the reported gains. Could the authors provide more tightly matched comparisons against baselines using the same backbone, training data, and optimization setup, but without the proposed generative map formulation?

3. How much of the final performance improvement comes from Stage 2 reinforcement fine-tuning relative to Stage 1 supervised training? Please provide more detailed evidence on the contribution of RL, including training stability, variance across runs, and sensitivity to reward design or optimization details.

4. The real-world humanoid robot experiment is interesting, but currently qualitative. Could the authors provide a clearer evaluation protocol, such as the number of trials, success rate, failure cases, and comparison to a no-map or alternative baseline?

I would currently lean toward a borderline reject, but I would be open to increasing the rating if the rebuttal provides clarifications on the questions and concerns.

**Limitations:**

yes

**Strengths And Weaknesses:**

Strengths:

1. The paper addresses an interesting problem in vision-language navigation, namely how to learn spatial representations that are better aligned with downstream decision-making in continuous environments. This is a meaningful question for embodied AI, especially under monocular and partially observable settings.

2. The empirical performance is promising. The paper reports strong results on R2R-CE and RxR-CE in the monocular setting, and the inclusion of ablations on training stages, channel initialization, and token budgets improves the empirical coverage. The zero-shot robot deployment is also a useful qualitative demonstration of potential real-world applicability.

3. The paper is generally well organized and easy to follow.

Weaknesses:

1. The central claim of “task-driven” map learning is not fully convincing in its current form. In the first training stage, the model is supervised with manually designed BEV targets consisting of occupancy, distance, and landmark channels. These are already strong human-designed priors, so it remains unclear to what extent the final representation is genuinely task-driven rather than substantially shaped by the predefined supervision.

2. The empirical analysis does not fully isolate the source of the reported gains. While the paper compares against prior methods and includes several ablations, it does not provide enough tightly matched comparisons to show that the improvements specifically come from the proposed generative map formulation, rather than other design or training choices.

3. The reinforcement learning stage is potentially important to the paper’s main claim, but its contribution is not analyzed deeply enough. In particular, the paper would benefit from clearer evidence about how much RL contributes beyond supervised pre-training, how stable training is, and how sensitive the method is to reward design and optimization details.

4. The real-world evidence is still limited. The humanoid robot example is interesting, but it remains qualitative and anecdotal. Without a quantitative protocol, repeated trials, or baseline comparisons, it is difficult to assess how strong the real-world generalization claim actually is.

---

> ### Author Rebuttal · Authors · 2026-03-31
>
> Thanks for recognizing our spatial representations for embodied AI, strong results, and real-world robot deployment. We address your concerns below.
>
> **Q1. Clarify "Task-Driven" Claim**
>
> Thanks for the question. We agree Stage 1 introduces human priors via manual BEV supervision. However, "task-driven" emphasizes that the **final** representation is ultimately optimized for downstream navigation objectives, not rigidly constrained by Stage-1 targets.
>
> The key evidence is twofold: Stage 2 narrows the gap to the full setting, and the final results become very similar across different initializations. In **Table 4 of the original submission**, reinforcement fine-tuning halves the average single-channel gap: **SR (2.5 to 1.3), SPL (1.6 to 0.8)**. Final Stage-2 results converge: Distance (44.9 SR/40.0 SPL), Landmark (44.5/39.7), and Occupancy (43.6/39.4), all closely approaching the full setting (45.6/40.5). Thus, Stage 1 provides a structured prior, while Stage 2 optimizes the representation for navigation, minimizing reliance on specific handcrafted targets.
>
> **Q2. Provide Tightly Matched Comparisons**
>
> Thanks. Due to the limited rebuttal period, we isolate the gains via tighter comparisons against representative SOTAs to clarify that improvements stem from our generative map, not other training choices.
>
> In Table A, **using the same setup** (backbone, data, and optimization), MapDream outperforms others in the same setting, confirming our gains stem directly from the generative map formulation.
>
> *Table A. Strict comparison for the source of performance gains (Trained on R2R-CE).*
> |Method|Backbone|Training Data|Batch Size|Epoch|lr|NE↓|OSR↑|SR↑|SPL↑|
> |:-|:-|:-|:-|:-|:-|:-|:-|:-|:-|
> |Baseline(w/o Map)|NVILA-2B|R2R-CE|64|1|1e-5|7.52|43.1|37.7|32.1|
> |AuxThink|NVILA-2B|R2R-CE|64|1|1e-5|7.15|45.3|40.4|34.2|
> |MonoDream|NVILA-2B|R2R-CE|64|1|1e-5|6.76|48.9|44.8|38.1|
> |MapDream(Ours)|NVILA-2B|R2R-CE|64|1|1e-5|**6.35**|**51.3**|**45.6**|**40.5**|
>
> **Q3. Provide Detailed Evidence on the Contribution of RL**
>
> Thanks. We provide multiple evidences to clarify the contribution of Stage 2 RL.
>
> **Training Stability and Variance Across Runs**
>
> Training is highly stable across configurations. **Our results in original paper were already averaged over three independent runs and we add standard deviations in Table B**.
> Stage 2 (RFT) improves average performance and reduces variance (**45.6%(±0.2%) SR**).
>
> *Table B. Training Stability (Trained on R2R-CE).*
>
> |Map|SPT|RFT|NE↓|OSR↑|SR↑|SPL↑|
> |:-|:-|:-|:-|:-|:-|:-|
> |-|-|-|7.67±0.08|45.8±0.6|37.7±0.5|32.1±0.5|
> |✓|✓|-|7.03±0.05|48.0±0.4|42.2±0.3|36.5±0.4|
> |✓|✓|✓|**6.35±0.04**|**51.3±0.2**|**45.6±0.2**|**40.5±0.3**|
>
> **Sensitivity to Optimization Details**
>
> Our method is highly robust to optimization details. Varying learning rates and GRPO rollout counts yields consistent metrics (Tables C1 and C2). SR remains **59.0~59.8**, and SPL stays **above 53.0**, confirming strong stability.
>
> *Table C1: Impact of learning rate (Trained on R2R-CE and RxR-CE).*
>
> |Exp.|#Rollout|LR|NE↓|OSR↑|SR↑|SPL↑|
> |:-|:-|:-|:-|:-|:-|:-|
> |1|8|1e-4|4.88|62.7|59.0|53.3|
> |2|8|5e-5|4.66|64.2|59.4|54.1|
> |3|8|1e-5|4.61|64.3|59.8|54.3|
>
> *Table C2: Impact of rollout count (Trained on R2R-CE and RxR-CE).*
>
> |Exp.|#Rollout|LR|NE↓|OSR↑|SR↑|SPL↑|
> |:-|:-|:-|:-|:-|:-|:-|
> |1|4|1e-5|4.77|63.9|59.1|54.0|
> |2|8|1e-5|4.61|64.3|59.8|54.3|
>
> **Sensitivity to reward design**
>
> Our reward function uses a **minimal yet essential set** of format and action rewards for structured reasoning and task grounding. Removing the format reward collapses outputs into unparsable text. Removing the action reward eliminates goal-directed signals, making navigation unlearnable.
>
> **Q4. Provide a Clearer Evaluation Protocol**
>
> Thanks. Our core contributions of the map formulation and two-stage training are effectively evaluated in simulation following the previous SOTAs. Moreover, the original real-world demos intuitively showcase the capability of our method to handle complex physical environments.
>
> To strengthen real-world evidence during the rebuttal period, we add new quantitative evaluations using **23 instructions**, repeated trials, and a unified protocol, as a controlled, small-scale supplement to our original demo.
>
> In Table D, MapDream achieves a much higher SR, proving our learned map representation benefits real-world deployment.
>
> *Table D. Quantitative Evaluation in Real World.*
> |Method|Instructions|Successes|SR↑|
> |:-|:-|:-|:-|
> |Baseline(No-map)|23|6|26.1|
> |MapDream(Ours)|23|11|47.8|
>
> **Analysis of Failure Cases**
>
> In the **5 cases where only MapDream succeeded**, one instruction was: “Walk straight down the hallway and turn left at the open doorway. Continue down the corridor and turn left at the end, then stop in front of the first cardboard box.” The baseline turned incorrectly and missed the target, whereas MapDream executed both turns and stopped accurately. Other baseline failures similarly involved wrong turns or overshooting.

---

> > ### Author Rebuttal · Reviewer_xqed · 2026-04-02
> >
> > Thanks to the authors for the detailed rebuttal and the new experiments. I believe most of my concerns have been addressed.
> >
> > I recommend incorporating these materials into the final version (main paper or appendix) to make the contribution more complete.
> >
> > I will increase my score for weak acceptance.

---

### Official Review · Reviewer_pgNA · 2026-03-11

**Soundness:** 3
**Presentation:** 2
**Significance:** 2
**Originality:** 3
**Overall Recommendation:** 4
**Confidence:** 2

**Summary:**

This paper proposes MapDream, a novel "map-in-the-loop" framework for Vision-Language Navigation (VLN) in continuous environments. Unlike conventional approaches that treat map construction as an exhaustive, task-agnostic 3D reconstruction problem, MapDream formulates it as an autoregressive bird's-eye-view (BEV) image synthesis process. The framework utilizes a two-stage optimization pipeline: a supervised pre-training stage to bootstrap the mapping-to-control interface, followed by a joint reinforcement fine-tuning (GRPO) stage. This design enables the map representation to be directly shaped by downstream navigation objectives. Empirically, the model achieves state-of-the-art results among monocular methods on R2R-CE and RxR-CE benchmarks, and demonstrates zero-shot sim-to-real transfer capabilities on a physical humanoid robot.

**Compliance With Llm Reviewing Policy:**

Affirmed.

**Key Questions For Authors:**

1. Global Mapping: Given that the generated BEV maps are egocentric and localized, how does the agent recover from extreme visual aliasing or perform deep backtracking when the required spatial context falls outside the historical observation window?
2. Temporal Stability: Are there any implicit mechanisms preventing severe topological flickering in the generated BEV maps between consecutive frames (t and t+1), especially under abrupt camera rotations?
3. Generalization of Pre-training: How sensitive is the framework to the quality of the IRef-VLA annotations during Stage 1? Could the system still cold-start effectively if only sparse heuristic labels were provided?

**Limitations:**

Yes.

**Strengths And Weaknesses:**

- Strengths: The conceptual shift from "mapping for reconstruction" to "mapping for decision-making" is highly innovative. By simultaneously training the map generator and the VLN policy under a unified navigation loss during the reinforcement fine-tuning stage, the framework enables end-to-end joint optimization. This elegantly solves the semantic mismatch between spatial representation and control by allowing the map to be directly shaped by downstream tasks. Significance (Compact 3-Channel Representation): Instead of exhaustive environment reconstruction, the method distills complex environmental context into a compact three-channel BEV map (Occupancy, Distance, and Landmark) that preserves only navigation-critical affordances. This efficient representation translates to impressive empirical results, particularly a dominating Success Weighted by Path Length (SPL) of 54.4 on R2R-CE, and enables zero-shot sim-to-real deployment on a Unitree G1 robot. Soundness: The dual-stage optimization is methodologically rigorous. Extensive ablation studies convincingly demonstrate the framework's robustness to map channel initializations and its computational efficiency at extremely low resolutions (96x96). Presentation: The paper is well-structured, with a clear narrative explaining the paradigm shift. The visual diagrams (especially the architecture overview and qualitative trajectory comparisons) effectively communicate the core mechanisms

- Weaknesses: The current architecture relies on a local, sliding-window egocentric BEV representation. It lacks explicit global spatial memory modeling, which may lead to "positional amnesia" in extremely long-horizon tasks involving complex backtracking. Soundness (Temporal Consistency): The autoregressive generation of maps at each independent time step lacks explicit geometric constraints for temporal consistency. Rapid camera shifts could potentially induce topological flickering in the BEV maps. Methodology Dependency: The supervised pre-training phase relies heavily on dense 3D referential grounding labels (e.g., IRef-VLA), which might pose a cold-start bottleneck when adapting to entirely novel, unlabeled domains.

---

> ### Author Rebuttal · Authors · 2026-03-31
>
> Thank Reviewer pgNA for the thoughtful review. We are encouraged by your appreciation of our paper's clarity and your support for our decision-centric mapping and optimization pipeline. We welcome this opportunity to discuss global Mapping, consistency, and cold-start issues in detail.
>
> **Q1. Add Analysis on Global Mapping**
>
> We thank Reviewer pgNA for this question. We would like to first clarify that global mapping or long-horizon global memory is not the main claim or key innovation of our current work. Our method is designed around a local ego-centric BEV formulation, and the manuscript does not claim that this alone fully addresses the broader open problem of global spatial memory. We therefore view the reviewer’s comment as pointing to an important adjacent direction, rather than a contradiction to our current scope.​
>
> Within our current local-BEV formulation, we have already examined the effect of spatial coverage by varying the map scale, which is also relevant to the reviewer’s question. To make this comparison meaningful, we used GT maps to remove the confounding effect of map generation quality, so that the results reflect the effect of spatial scale itself rather than differences in synthesized map accuracy. Concretely, with the resolution fixed at 448×448, we evaluated three physical scales: 1.25 cm/px, 2.5 cm/px, and 5.0 cm/px.
>
> ​The results show that **larger coverage does not automatically lead to better performance**. In particular, 2.5 cm/px achieves the best overall result (SR 90.2, SPL 87.0), while a larger map at 5.0 cm/px performs worse (SR 86.1, SPL 82.6). This suggests that, under a fixed resolution, simply enlarging the local map can introduce more redundant or diluted spatial information, which may be less helpful for immediate action prediction.
>
> ​At the same time, we do not view this result as evidence that broader spatial context is unimportant. Rather, it suggests that naively enlarging a local BEV window may be insufficient, and that better mechanisms for long-horizon memory or multi-scale/global information integration may be needed. In this sense, we believe **the reviewer’s intuition is valuable**, and our result mainly indicates that this direction likely **requires a more principled design** than a straightforward increase of local map coverage.
>
> *Table A : Ablation on the spatial scale of the local map (Stage 1 with GT maps on R2R; Trained on R2R-CE).*
>
> | Scale      | NE ↓     | OSR ↑    | SR ↑     | SPL ↑    |
> | :--------- | :------- | :------- | :------- | :------- |
> | 5.0 cm/px  | 1.71     | 87.7     | 86.1     | 82.6     |
> | 1.25 cm/px | 1.37     | 89.5     | 89.5     | **87.6** |
> | 2.5 cm/px  | **1.33** | **90.7** | **90.2** | 87.0     |
>
> **Q2. Is severe topological flickering implicitly mitigated?**
>
> Yes, to some extent. The main implicit mechanism is that the map generator is conditioned on temporally correlated observation histories, rather than predicting each BEV map from an isolated frame. Each prediction is conditioned on the full observation history up to the current step, so two consecutive predictions share all previous observations and differ only by one newly added frame. This provides short-term temporal continuity and makes abrupt map changes less likely unless the new observation brings sufficiently strong new visual evidence.
>
> ​We will clarify this source of implicit temporal stability more explicitly in the revised manuscript.
>
> **Q3. Analyze Sensitivity to Stage-1 supervision**
>
> Our framework is not strongly dependent on the dense IRef-VLA annotations, which in our design correspond to the Landmark channel. This is already supported by **Table 4 in the original submission**. The other two channels, Distance and Occupancy, do not use IRef-VLA annotations, yet they still achieve competitive performance.
>
> ​Specifically, with Stage-1 supervised pre-training only, Distance reaches 39.4 SR / 34.5 SPL and Occupancy reaches 39.4 SR / 34.8 SPL, compared with 40.4 SR / 35.3 SPL for Landmark and 42.2 SR / 36.5 SPL for the full setting. After Stage-2 reinforcement fine-tuning, **Distance further reaches 44.9 SR / 40.0 SPL** and Occupancy reaches 43.6 SR / 39.4 SPL, which remain **close to Landmark (44.5 SR / 39.7 SPL)** and the full setting (45.6 SR / 40.5 SPL).
>
> ​These results suggest that dense IRef-VLA annotations are helpful but not necessary for cold start. Even without the Landmark channel, sparse heuristic supervision alone is sufficient to initialize a strong model, which is then further improved in Stage 2.

---

> > ### Author Rebuttal · Reviewer_pgNA · 2026-04-03
> >
> > I appreciate the authors’ detailed response and the additional analyses provided in the rebuttal. The clarifications help improve the clarity of the work and address several of my questions.
> > Overall, my assessment of the paper remains unchanged. （4）

---

### Official Review · Reviewer_fV2r · 2026-03-16

**Soundness:** 3
**Presentation:** 3
**Significance:** 3
**Originality:** 2
**Overall Recommendation:** 4
**Confidence:** 4

**Summary:**

MapDream is an approach to vision-and-language navigation (VLN) that learns to build a top-down map while navigating from natural language instructions. The core idea is that the map should be a learned representation shaped by the needs of the navigation objective rather than try to reconstruct everything faithfully (when much of the scene detail is potentially irrelevant to navigation). In essence, the paper proposes that during training time gradients should flow to the map construction module (whereas in previous works, the map construction is learned separately, using auxiliary objectives, or based on classical approaches).

MapDream is trained in 2 stages. Supervised finetuning of the map module and the VLN policy separately, then RL finetuning of both components jointly. It achieves SOTA results on R2R-CE and RxR-CE (continuous environment versions of the tasks).

**Compliance With Llm Reviewing Policy:**

Affirmed.

**Final Justification:**

The detailed rebuttal addressed my concerns, and leaves me comfortable retaining the weak accept rating. The rating was already assuming my questions could be addressed.

**Key Questions For Authors:**

See above

**Limitations:**

yes

**Strengths And Weaknesses:**

Strengths
- Interesting framing - reframing map construction as image synthesis and putting it inside the learning loop
- Two stage training makes sense
- Strong ablations - Table 4 - really cool to see that RL narrows the gap between different channel ablations, showing that it is working as intended. Also Table 5.
- fantastic to see that it was actually deployed on a robot

Weaknesses
- The "task-driven" claim might be a bit overstated, given that the supervision in stage 1 is handcrafted expert-designed signals (e.g. geodesic distance to goal, landmarks from labels).
- I don't understand why the full setting in Table 3 doesn't match the results in Table 5. Can the authors please explain this.
- I would be interested to see StreamVLN and JanusVLN included in the results table. This does not harm the papers chances (I understand JanusVLN is concurrent work at ICLR 2026 and StreamVLN is unpublished) but it just makes it easier to track the field.
- There is no detailed understanding of how the map is used by the model apart from quantitative improvement on final numbers. For example, I wonder what would be the upper bound if the agent started with the ground truth map?

Minor weaknesses
- Format reward (eq 5) seems trivial but gets a lot of space
- How does the gradient flow back through the discrete BEV images tokens to the map module? I don't think this is discussed.

Overall:
- In my opinion this is a solid paper with strong ablations and very good results. I think it would be of interest to the community and I vote for acceptance, assuming my questions above can be satisfactorily answered.

---

> ### Author Rebuttal · Authors · 2026-03-31
>
> We thank Reviewer fV2r for recognizing our learning-based map formulation, two-stage training, comprehensive ablations, and real-world deployment. We clarify your key questions below.
>
> **Q1. Clarify the "Task-Driven" Claim**
>
> We thank Reviewer fV2r for this question. We clarify that “task-driven” refers to the **final objective** that shapes the learned representation, rather than the absence of initial supervision. In our framework, Stage 1 serves only as a structured cold start to bootstrap usable BEV map generation, without determining the final representation.
>
> Importantly, **as shown in Table 4 of the original submission**, Stage 2 significantly reduces the gap caused by different Stage 1 supervision targets. Compared with the full-channel setting, the average gap of single-channel initialization **decreases from 2.5 to 1.3 on SR, from 1.6 to 0.8 on SPL, and from 2.0 to 0.8 on OSR** after Stage 2. This indicates that the method is not strongly dependent on any specific handcrafted Stage 1 target (e.g., Distance, Landmark, Occupancy), while Stage 1 mainly provides a consistent BEV output format.
>
> We will follow the reviewer’s suggestion and clarify this point more explicitly in the revised manuscript.
>
> **Q2. Explain Why Table 3 Doesn't Match Table 5**
>
> We thank Reviewer fV2r for this observation. The discrepancy arises from different training stages: Table 5 reports Stage-1-only models (supervised pre-training), while Table 3 reports the final model after both Stage 1 and Stage 2 (reinforcement fine-tuning). As stated in the paper, Table 5 is designed to evaluate the impact of reduced BEV spatial resolution on performance.
>
> We perform this ablation only in Stage 1 because higher-resolution BEV maps significantly increase inference cost and slow down Stage 2. Increasing the resolution from 96×96 to 384×384 raises tokens from 36 to 576 and inference time **from 1.3s to 12.7s (nearly 10×)**. Since Stage 2 repeatedly calls map generation, this overhead directly slows end-to-end training. Therefore, we restrict this ablation to Stage 1.
>
> **Q3. Include StreamVLN and JanusVLN in the Results Table**
>
> We thank Reviewer fV2r for this suggestion. We agree that including recent concurrent works such as StreamVLN and JanusVLN would make the comparison more complete and better reflect the evolving VLN landscape. As these works were not publicly available at submission time, we will include them in the revised manuscript (Table 1).
>
> **Q4. Analyze How the Map is Used and Upper Bound with Ground Truth Map**
>
> We thank Reviewer fV2r for this question. We address it from two aspects: (1) how the map is used in our model, and (2) what an oracle upper bound with GT maps looks like.
>
> **How the Map is Used:** The generated map is concatenated with historical observations and the instruction to form a unified prompt, which serves as the context for predicting the next action. These prompts are described in the original paper (lines 177 and 193, right column).
>
> **Oracle upper bound with GT maps.:** To assess the utility of the map signal, we test an oracle setting replacing the generated map with a GT map. Performance improves from 42.2 SR / 36.5 SPL to **90.2 SR / 87.0 SPL**, showing that the map is highly informative and that better map quality substantially improves navigation.
>
> We note that this oracle result does not imply exact GT map reconstruction is the goal. **GT maps are privileged and unavailable at test time**. Our objective is to learn maps most useful for navigation, not necessarily geometrically identical to the ground truth. Thus, the GT result should be viewed only as an upper bound on the value of spatial map information.
>
> *Table A : Performance comparison evaluating the utility of map representations (Stage 1; Trained on R2R-CE).*
>
> |Setting|NE↓|OSR↑|SR↑|SPL↑|
> |:-|:-|:-|:-|:-|
> |w/o Map|7.67|45.8|37.7|32.1|
> |w/ Generated Map|7.03|48.0|42.2|36.5|
> |w/ GT Map|**1.33**|**90.7**|**90.2**|**87.0**|
>
> **Q5. Reduce Space for Format Reward (Eq 5)**
>
> We will simplify the format reward and reduce its space in the revised manuscript.
>
> **Q6. Discuss Gradient Flow Back Through Discrete BEV Images Tokens**
>
> We thank Reviewer fV2r for this question. Gradients to the map generator are not passed through discrete BEV tokens, but via the GRPO objective, where downstream navigation advantage weights the log-probabilities of sampled BEV tokens (map model) and actions (navigation model). That is, we do not differentiate through token discretization. Instead, after executing actions, we compute reward and advantage from the navigation outcome and use this shared advantage to jointly optimize both models.

---

> > ### Author Rebuttal · Reviewer_fV2r · 2026-04-05
> >
> > Thanks for the detailed rebuttal, it addresses my concerns making me feel comfortable retaining the weak accept rating.

---

### Decision · Program_Chairs · 2026-04-30

**Decision:**

Accept (regular)

**Comment:**

This paper introduces map learning as a BEV image prediction task for VLN. It jointly predicts maps and actions while operating in a partially observed space.

Overall, the reviews highlight the novelty in the map prediction being incorporated into the VLN learning task. The empirical results look compelling, and the ablations and baselines sufficient for the proposed approach.

The main concern raised across the reviews was the "task-driven" claim. Having read the reviews and the author response, I agree that this part may be a bit oversold. I suggest toning down the claim of being task-driven.

Overall, I believe the reviews support that this is a solid technical contribution.